# A Study on Analyses of the Production Data of Feed Crops and Vulnerability to Climate Impacts According to Climate Change in Republic of Korea

MoonSun Shin [1], Seonmin Hwang [1], Junghwan Kim [1], Byungcheol Kim [2] and Jeong-Sung Jung [3,*]

1   Department of Computer Engineering, Konkuk University, Chungju 27478, Republic of Korea; msshin@kku.ac.kr (M.S.); smhwang@kku.ac.kr (S.H.)
2   Department of Information and Communication, Baekseok University, Cheonan 31065, Republic of Korea; bckim@bu.ac.kr
3   Division of Grassland & Forage, National Institute of Animal Science, Cheonan 31000, Republic of Korea
*   Correspondence: jjs3873@korea.kr

**Abstract:** According to the climate change scenario, climate change in the Korean Peninsula is expected to worsen due to extreme temperatures, with effects such as rising average temperatures, heat waves, and droughts. In Republic of Korea, which relies on foreign countries for the supply of forage crops, a decrease in the productivity of forage crops is expected to cause increased damage to the domestic livestock industry. In this paper, to solve the issue of climate vulnerability for forage crops, we performed a study to predict the productivity of forage crops in relation to climate change. We surveyed and compiled not only forage crop production data from various regions, but also experimental cultivation production data over several years from reports of the Korea Institute of Animal Science and Technology. Then, we crawled related climate data from the Korea Meteorological Administration. Therefore, we were able to construct a basic database for forage crop production data and related climate data. Using the database, a production prediction model was implemented, applying a multivariate regression analysis and deep learning regression. The key factors were determined as a result of analyzing the changes in forage crop production due to climate change. Using the prediction model, it could be possible to forecast the shifting locations of suitable cultivation areas. As a result of our study, we were able to construct electromagnetic climate maps for forage crops in Republic of Korea. It can be used to present region-specific agricultural insights and guidelines for cultivation technology for forage crops against climate change.

**Keywords:** forage crop productivity; climate vulnerability; predictive model; suitable cultivation area; electromagnetic climate map

## 1. Introduction

Climate change has already emerged as a global concern, prompting significant efforts at the national level to address crisis situations arising across various fields due to changing climate conditions. In particular, the agricultural and livestock industries are greatly affected by climate change, resulting in the reduced production and quality of agricultural and livestock products. Therefore, there is a need to prepare and establish policies for cultivating sustainable agricultural and livestock products and creating conducive breeding environments at a national level. For agronomic management and improving agricultural production, much research has been performed, applying machine learning and deep learning [1]. Precisely monitoring the growth conditions and nutritional status of maize is crucial, and can be possible through the use of unmanned aerial vehicles (UAVs) [1]. A study on suggestions and policies that can improve our capabilities to reduce the damage caused by climate change was proposed [2]. The Korean government is increasing its climate change budget to address climate change issues in various industrial fields [2].

However, due to recent climate change, overall production and the movement of cultivation areas are changing. According to the climate change scenario Representative Concentration Pathway (RCP) 8.5, the Korean Peninsula is expected to experience increased climate change impacts, including higher average temperatures, heat waves, and droughts [3,4].

Figure 1 shows changes in temperature and precipitation in the Korean Peninsula according to the RCP scenario [4]. These abnormal temperatures are anticipated to significantly affect the agricultural sector, including forage crops [5]. Consequently, the incidence of weather-related disasters and damage due to abnormal weather is on the rise. If the average temperature increases by more than 3 °C due to climate change, crop yields will decrease in all regions of the world [6]. In particular, the production of coarse forage is expected to decrease by more than 25%. Given the limited production and cultivation area for coarse forage in Korea, the global decrease in feed supply will serve as a significant factor placing a burden on livestock farmers [7]. In Republic of Korea, a country that depends on foreign sources for its supply of forage crops, the projected decline in forage crop productivity is expected to exacerbate the impact of climate change on the domestic livestock industry [8].

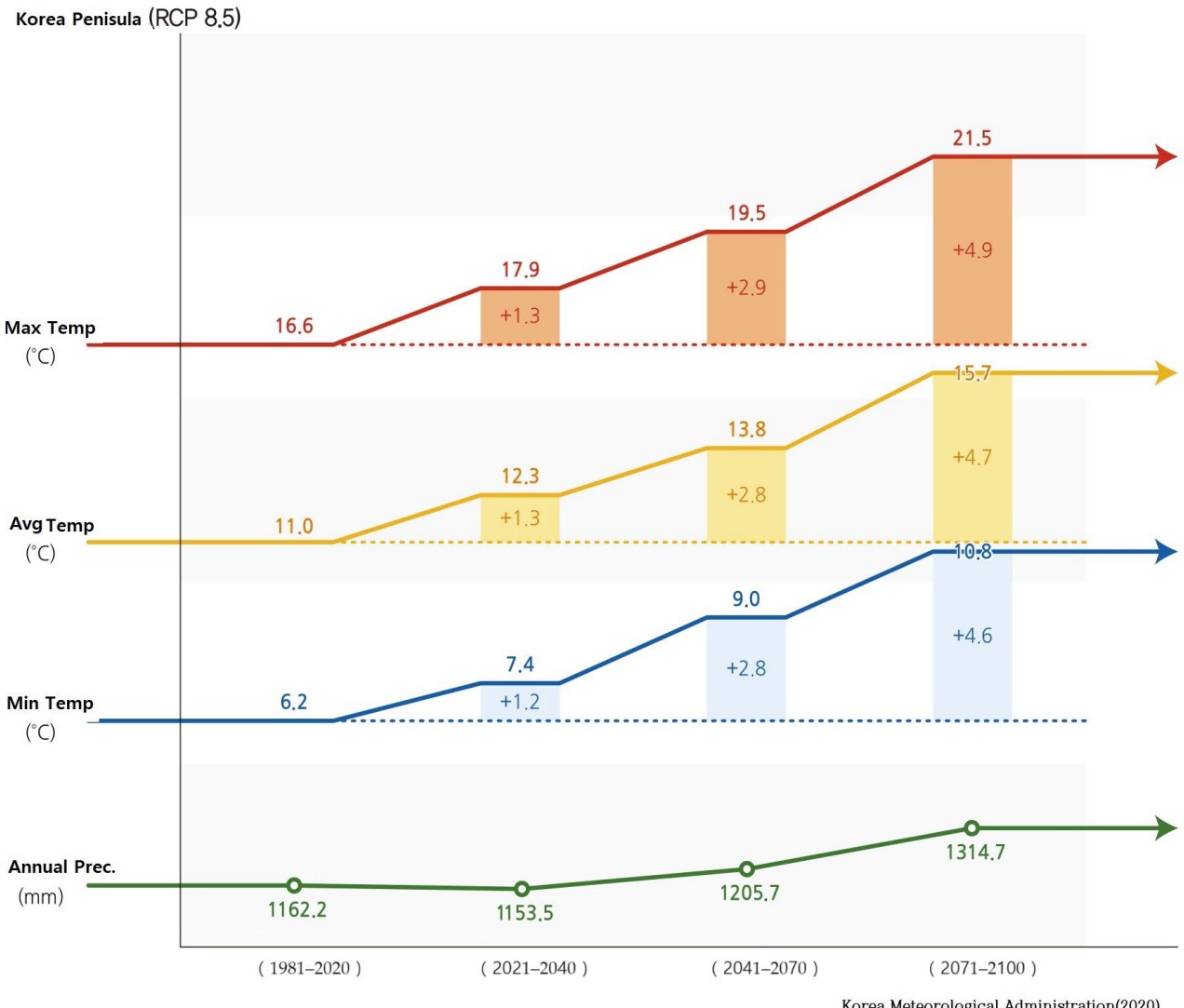

**Figure 1.** Climate change on the Korean Peninsula (RCP 8.5).

Due to climate change, the temperature in Republic of Korea has risen by 1.5 °C, much higher than the global average rise over the past 100 years, changing the suitable

cultivation limit for crops [9–12]. This means that the cultivation area for grasses and forage crops is moving north from the southern region to the central region due to rising temperatures. Some surveys are underway to assess changes in the suitable cultivation limits, yields, and cultivation environments. It is necessary to evaluate the vulnerability to climate change [13,14]. Due to the ongoing climate warming, summer crops have been negatively affected due to high temperatures, while winter crops have been reported to increase productivity per area due to the increasing temperature during the wintering and growing stages [7].

The objective of our paper was to predict changes in the cultivation area and productivity of forage crops in Korea in response to climate change in the Korean Peninsula. Therefore, we performed to create an electromagnetic climate map for identifying suitable cultivation areas by collecting production data for forage crops, establishing correlations with climate data, analyzing vulnerabilities to specific climate factors, tracking changes in the boundaries of suitable cultivation areas, and developing production prediction models for each forage crop based on climate factors. Although research has been performed on the production of many crops, including rice, wheat, barley, cabbage, and radish crops, not much research has been conducted on feed crops in Korea [5]. Recently, in order to improve the productivity of livestock farms, research on the productivity of feed crops is necessary, so the National Institute of Animal Science conducted a survey on production data for each feed crop and studied the experimental results. A major contribution of this study is that we collected production data through various sources, collected climate data according to the region and year of the production data, and established a database related to feed crops. The existing approach was to experiment with soil, fertilizer, etc. [3], but in this study, data were analyzed to determine the relationship between forage crop production and climate impact, and an electronic climate map of changes in cultivation areas that could be provided to livestock farms was constructed.

The rest of this paper is organized as follows: In Section 2, the basic data collection and preprocessing are described and the research methodology is presented. The results of the analysis of the climate impact vulnerability of forage crops and the construction of a prediction model for forage crop production are described in Section 3. In addition, experiments to improve the coefficient of determination of the models are presented. Section 4 presents the electromagnetic climate map for the changes in the suitable cultivation area. Finally, conclusions and future works are described in Section 5.

## 2. Data Collection and Preprocessing

In this section, the research methodology is described, and the process of building a basic database, including data collection, are presented. Our research methodology is described below. The goal of our study is to predict forage crop production and build an electronic climate map according to climate change.

A framework of the research methodology process is shown in Figure 2. The flow process is as follows: (1) the collection of pasture/fodder crop production data, (2) the collection of climate data according to the year and region of the collected production data, (3) the selection of the most influential climate elements in response to the production data, (4) the creation of various regression models and a comparison analysis of the results, (5) the establishment of a production prediction model with climate factors that affect each feed crop, and (6) the construction of an electronic climate map, accordingly. This can serve as basis for the movement of suitable cultivation areas for feed crops due to climate change in the Korean Peninsula, and is planned to be used for farming guidance for livestock farmers in Republic of Korea.

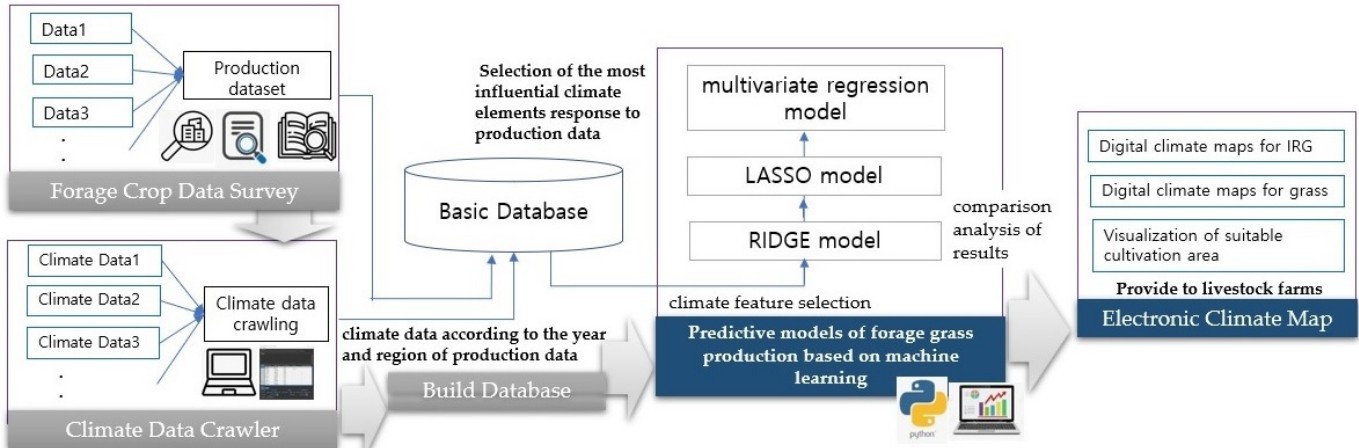

**Figure 2.** A framework of the research methodology process.

To analyze the impact of climate on grasses and forage crops resulting from climate change, to predict production, and to figure out shifts in the suitable cultivation areas, it is necessary to analyze forage crop production data and climate data. While the IRG data and forage data were collected through literature reviews and experimental reports from the Korea Institute of Animal Science and Technology, for climate data collection, we developed a climate data crawler using Python 3.10.7. This study was conducted in collaboration with the Korea Institute of Animal Science and Technology. To assess the vulnerability of forage crops to climate change, production data, with labels of the region and year as referencing labels, as well as the climate data of the corresponding region and year could be crawled.

The climate data crawler implemented using Python carries out to crawl the climate data obtained from the Korea Meteorological Administration's Automatic Weather Station (AWS) and synoptic weather observation system, the Automated Surface Observing System (ASOS). The procedures to collect the climate data were as follows: First, the type of climate data and the date of the climate data to be collected were selected from the Start menu of the climate data crawler. After that, a region in the 'Configuration' tab was selected and the 'Start' button for crawling was clicked. Climate data for the chosen region and year were then collected. The data were saved as a CSV file. The collected climate data and forage crop production data were merged to establish a basic dataset, which was necessary for the analysis.

Recently, there has been a growing interest in Italian ryegrass as a forage crop that can be cultivated in Republic of Korea [3,13]. Italian ryegrass (IRG) is a representative winter forage crop and can be harvested from late April to May of the following year after sowing rice in Republic of Korea. In particular, it is resistant to injury via moisture, so it is possible to cultivate paddy fields after harvesting rice [3].

Italian ryegrass has an excellent livestock palatability, a high crude protein content, and a high digestible nutrient content. Thus, it is very valuable as a feed substitute for assorted feed. Its quality of silage is also very good due to its high sugar content compared to feeding rice straw [9,14]. In the case of dairy cows, while feeding Italian ryegrass can lead to increasing milk production, higher milk fat, and higher milk protein content, it has been proven to consequently improve milk quality [15–17]. However, in recent times, due to abnormal climate conditions, the suitable cultivation areas of Italian ryegrass (IRG) are shifting and the pace of these changes is accelerating [18–20].

The purpose of this study was to analyze the vulnerability of Italian ryegrass, a winter forage crop, and grasses to climate change, and to determine the limits of the suitable cultivation areas. Thus, data preprocessing was conducted to analyze the production data of IRG and other grasses, which were surveyed by the National Institute of Animal Science. Climate data corresponding to the respective regions and years of each production dataset

were collected, resulting in the establishment of a comprehensive database integrating the production data into a basic dataset.

For Italian ryegrass, the first dry matter yield was established as the dependent variable. Similarly, for the grass data, where the overall dry matter yield held significance, the total dry matter yield was designated as the dependent variable. Following this, several climate factors were chosen as independent variables. An analysis was then performed. Figure 3 shows the basic dataset DB that was constructed after the preprocessing of the IRG data.

| DMY1 | PREOCT | PRESUM | PREAFOVWIN | MINTDEC | MINTJAN | MINTFEB | MINTMAR | GDDFJTA |
|---|---|---|---|---|---|---|---|---|
| 6,652 | 73.8 | 401.4 | 73.8 | -3.9 | -8.2 | -3.8 | 0.6 | 293.05 |
| 6,476 | 73.8 | 406.4 | 73.8 | -3.9 | -8.2 | -3.8 | 0.6 | 293.05 |
| 13,125 | 47 | 342.6 | 59.9 | -6.3 | -11.4 | -5.7 | -1.5 | 239.75 |
| 10,131 | 73.8 | 401.4 | 73.8 | -3.9 | -8.2 | -3.8 | 0.6 | 326.95 |
| 9,498 | 73.8 | 283.2 | 73.8 | -3.9 | -8.2 | -3.8 | 0.6 | 326.95 |
| 13,701 | 45.5 | 585.3 | 87.2 | -7.5 | -10.7 | -6.3 | -3 | 218.75 |
| 7,264 | 2 | 282.1 | 45.9 | -2.7 | -6.6 | -6 | -0.7 | 295.1 |
| 8,082 | 3.5 | 297.2 | 65.7 | -4.1 | -7.8 | -6.2 | -3.2 | 301.45 |
| 4,392 | 2 | 462.5 | 45.9 | -2.7 | -6.6 | -6 | -0.7 | 295.1 |
| 2,263 | 4.7 | 544.7 | 46.8 | -6.5 | -11.9 | -10 | -3.8 | 238.45 |
| 4,720 | 2 | 465 | 45.9 | -2.7 | -6.6 | -6 | -0.7 | 295.1 |
| 5,513 | 4.7 | 524.2 | 46.8 | -6.5 | -11.9 | -10 | -3.8 | 238.45 |
| 7,213 | 26.7 | 307.5 | 55 | -10.4 | -5.3 | 4.9 | -2.6 | 209.55 |
| 11,273 | 8.5 | 439.8 | 209.4 | 11.1 | 3.5 | 5.1 | 3.5 | 460.65 |
| 3,565 | 14.5 | 351.8 | 48.6 | -12.3 | -8.2 | -7.5 | -3 | 204.6 |
| 6,193 | 0.4 | 253 | 84 | -6 | -2.5 | -3.2 | 0.6 | 348.9 |
| 7,021 | 70.2 | 392.4 | 70.1 | -8.2 | -4.3 | -4.5 | 0.1 | 261.3 |
| 9,246 | 12 | 318 | 112 | -9 | -4.4 | -4.6 | -1.4 | 723.65 |
| 10,330 | 26.7 | 307.5 | 55 | -10.4 | -5.3 | 4.9 | -2.6 | 209.55 |
| 12,605 | 12 | 318 | 112 | -9 | -4.4 | -4.6 | -1.4 | 723.65 |
| 7,213 | 70.2 | 307.5 | 55 | -8.2 | -4.3 | -4.5 | 0.1 | 261.3 |
| 3,565 | 14.5 | 351.8 | 48.6 | -12.3 | -8.2 | -7.5 | -3 | 204.6 |

**Figure 3.** Preprocessed dataset of IRG.

In the process of data preprocessing, missing values and cases where climate data could not be obtained were all removed. Using the fundamental dataset DB created in this manner, variations in the productivity of grasses and forage crops in response to various climate factors were investigated. The preprocessed dataset of the grasses is shown in Figure 4.

To analyze the vulnerability to climate impacts, the following research methodology was used: Initially, a database (DB) was set up to track the variations in productivity of grasses and forage crops, followed by an analysis of these productivity changes. While there are numerous factors influencing crop productivity, in this paper, we have focused solely on climate factors to analyze changes in production and the cultivation area. Furthermore, this paper presents research results concerning Italian ryegrass, a winter forage crop, and other grasses, including corn, one of the most widely cultivated forage crops in Republic of Korea.

| DMYT | GDDTOTAL | DDAYS | MAXTJUL | MAXTAUG | PREDAUG | GDDFJTA | PREDNOV | MINTJAN | MINTFEB | MINTMAR | MAXTNOV | MAXTMAR | MTJAN | MTFEB | MTMAR |
|---|---|---|---|---|---|---|---|---|---|---|---|---|---|---|---|
| 14,976 | 3554.6 | 0 | 31.9 | 33.3 | 9 | 307.95 | 9 | -18.2 | -10.2 | -6.3 | 22.4 | 23.1 | -3 | 2.5 | 6.1 |
| 12,841 | 3554.6 | 0 | 31.9 | 33.3 | 9 | 307.95 | 9 | -18.2 | -10.2 | -6.3 | 22.4 | 23.1 | -3 | 2.5 | 6.1 |
| 14,771 | 3554.6 | 0 | 31.9 | 33.3 | 9 | 307.95 | 9 | -18.2 | -10.2 | -6.3 | 22.4 | 23.1 | -3 | 2.5 | 6.1 |
| 14,614 | 3554.6 | 0 | 31.9 | 33.3 | 9 | 307.95 | 9 | -18.2 | -10.2 | -6.3 | 22.4 | 23.1 | -3 | 2.5 | 6.1 |
| 13,289 | 3554.6 | 0 | 31.9 | 33.3 | 9 | 307.95 | 9 | -18.2 | -10.2 | -6.3 | 22.4 | 23.1 | -3 | 2.5 | 6.1 |
| 11,595 | 2899.45 | 9 | 30 | 31.6 | 13 | 214.05 | 14 | -18.9 | -13.4 | -8.8 | 18.2 | 20.9 | -6.4 | 0.3 | 3.9 |
| 13,533 | 2899.45 | 9 | 30 | 31.6 | 13 | 214.05 | 14 | -18.9 | -13.4 | -8.8 | 18.2 | 20.9 | -6.4 | 0.3 | 3.9 |
| 12,960 | 2899.45 | 9 | 30 | 31.6 | 13 | 214.05 | 14 | -18.9 | -13.4 | -8.8 | 18.2 | 20.9 | -6.4 | 0.3 | 3.9 |
| 13,064 | 2899.45 | 9 | 30 | 31.6 | 13 | 214.05 | 14 | -18.9 | -13.4 | -8.8 | 18.2 | 20.9 | -6.4 | 0.3 | 3.9 |
| 11,218 | 2899.45 | 9 | 30 | 31.6 | 13 | 214.05 | 14 | -18.9 | -13.4 | -8.8 | 18.2 | 20.9 | -6.4 | 0.3 | 3.9 |
| 14,350 | 3801.05 | 0 | 32.2 | 33.1 | 12 | 446.2 | 5 | -14.1 | -7.9 | -5 | 24.6 | 22.8 | -0.4 | 5.9 | 8.7 |
| 15,030 | 3801.05 | 0 | 32.2 | 33.1 | 12 | 446.2 | 5 | -14.1 | -7.9 | -5 | 24.6 | 22.8 | -0.4 | 5.9 | 8.7 |
| 13,690 | 3801.05 | 0 | 32.2 | 33.1 | 12 | 446.2 | 5 | -14.1 | -7.9 | -5 | 24.6 | 22.8 | -0.4 | 5.9 | 8.7 |
| 14,230 | 3801.05 | 0 | 32.2 | 33.1 | 12 | 446.2 | 5 | -14.1 | -7.9 | -5 | 24.6 | 22.8 | -0.4 | 5.9 | 8.7 |
| 13,920 | 3801.05 | 0 | 32.2 | 33.1 | 12 | 446.2 | 5 | -14.1 | -7.9 | -5 | 24.6 | 22.8 | -0.4 | 5.9 | 8.7 |
| 15,566 | 3554.6 | 0 | 31.9 | 33.3 | 9 | 307.95 | 9 | -18.2 | -10.2 | -6.3 | 22.4 | 23.1 | -3 | 2.5 | 6.1 |
| 15,589 | 3554.6 | 0 | 31.9 | 33.3 | 9 | 307.95 | 9 | -18.2 | -10.2 | -6.3 | 22.4 | 23.1 | -3 | 2.5 | 6.1 |
| 15,004 | 3554.6 | 0 | 31.9 | 33.3 | 9 | 307.95 | 9 | -18.2 | -10.2 | -6.3 | 22.4 | 23.1 | -3 | 2.5 | 6.1 |
| 14,295 | 3554.6 | 0 | 31.9 | 33.3 | 9 | 307.95 | 9 | -18.2 | -10.2 | -6.3 | 22.4 | 23.1 | -3 | 2.5 | 6.1 |
| 15,662 | 3554.6 | 0 | 31.9 | 33.3 | 9 | 307.95 | 9 | -18.2 | -10.2 | -6.3 | 22.4 | 23.1 | -3 | 2.5 | 6.1 |
| 14,415 | 3554.6 | 0 | 31.9 | 33.3 | 9 | 307.95 | 9 | -18.2 | -10.2 | -6.3 | 22.4 | 23.1 | -3 | 2.5 | 6.1 |
| 16,140 | 2899.45 | 9 | 30 | 31.6 | 13 | 214.05 | 14 | -18.9 | -13.4 | -8.8 | 18.2 | 20.9 | -6.4 | 0.3 | 3.9 |

**Figure 4.** Preprocessed dataset of forage grass.

After analyzing the productivity changes for both grasses and forage crops, an electromagnetic climate map was constructed. This map was based on an analysis of the criteria for determining suitable cultivation areas for these crops, while taking climate conditions and their variations into account. To analyze the impact of climate change and abnormal weather on the productivity of grasses and forage crops, a prediction model for their productivity was established and their vulnerability to climate change was analyzed. Furthermore, an electromagnetic model was developed based on the productivity prediction model.

## 3. Climate-Related Production Prediction Model for Forage Crops

A regression analysis model was used for the predictive modeling of yield according to climate factors of Italian ryegrass, a winter forage crop, as well as cold-season grasses. A production prediction model was developed by employing a multiple regression analysis to assess the impact of various independent variables, including climate factors, on the dependent variable—the dry matter yield. A multiple regression analysis was carried out using SPSS 26.0. As part of deep learning regression, both the Lasso and Ridge models were applied. Since there was a problem of multicollinearity due to the correlation of the climate factors, the Lasso model was applied to solve this problem. Python was used for the Lasso model and the Ridge model. Since the results of the Lasso model and the Ridge model were almost similar, only the results of the Lasso model were presented for comparative analysis.

### 3.1. Italian Ryegrass Production Prediction Model

To analyze the correlation between Italian ryegrass and climate data, a climate change vulnerability analysis was performed with the selected eight climate factors known to affect the growth of winter forage crops. These selected climate factors have been empirically shown to impact the growth of winter forage crops. These factors were chosen by the Grassland Feed Department of the National Institute of Animal Science. Their selection was based on the department's extensive experience and knowledge, gained over many years of the trial cultivation of grassland forage crops with previously studied climate factors. Dry matter yield (DMY1) was chosen as the dependent variable. The following eight climate factors were selected as independent variables: October precipitation (PREOCT), the sum of precipitation (PRESUM), the precipitation sum over winter (PREAFTOVWIN), the minimum temperature in December (MINTDEC), the minimum temperature in January

(MINTJAN), the minimum temperature in February (MINTFEB), the minimum temperature in March (MINTMAR), and the GDD From January to April (GDDFJTA). These variables are listed in Table 1. As shown in Section 2, N = 304. The descriptive statistics for each variable are summarized in Table 1. These valuables satisfied the normality of the dry matter yield (DMY1), the dependent variable.

**Table 1.** Variables and descriptive statistics of IRG dataset.

| Category | Variable | Meaning | Mean | S.E. |
|---|---|---|---|---|
| Yield | DMY1 | First Dry Matter Yield of IRG | 9011.877 | 4295.5617 |
| Precipitation | PREOCT | Precipitation in October | 50.2601 | 69.8917 |
| | PRESUM | Sum of Precipitation | 365.834 | 117.2916 |
| | PREAFTOVWIN | Precipitation Sum Over Winter | 93.315 | 56.7692 |
| Temperature | MINTDEC | Minimum Temperature in December | −3.323 | 5.1069 |
| | MINTJAN | Minimum Temperature in January | −5.595 | 5.0707 |
| | MINTFEB | Minimum Temperature in February | −3.714 | 4.6233 |
| | MINTMAR | Minimum Temperature in March | 1.490 | 3.2148 |
| GDD | GDDFJTA | GDD From January To April | 339.207 | 133.038 |

Table 2 shows the results of the correlation analysis between the eight climate indicators and the dry matter yield. As a result of the correlation analysis between these eight climate factors and the dry matter yield, variables such as the growing degree days from January to April (GDDFJTA), the minimum temperature in January (MINTJAN), the precipitation in October (PREOCT), and the precipitation sum over winter (PREAFOVWIN) exhibited slightly high correlation coefficients. Additionally, it was evident that all independent variables were correlated with the dependent variable, the dry matter yield (DMY1). These correlations were statistically significant at the 0.01 level. Figure 5 shows a graph of the correlation analysis results for each variable.

**Table 2.** Results of correlation analysis between IRG production and 8 climate indicators.

| | | PREOCT | PRESUM | PREAFOVWIN | MINTDEC | MINTJAN | MINTFEB | MINTMAR | GDDFJTA |
|---|---|---|---|---|---|---|---|---|---|
| Pearson | DMY1 | 0.445 ** | 0.407 ** | 0.493 ** | 0.574 ** | 0.600 ** | 0.569 ** | 0.583 ** | 0.564 ** |

** The correlations are statistically significant at the 0.01 level.

To select the climate factors with more influence on Italian ryegrass production among the eight climate factors, forecast models were constructed and analyzed in various combinations. While developing a productivity prediction model, several variables, including the soil type, variety, and cultivation management, might exert an influence in addition to climate factors. However, since this study's objective was to analyze the vulnerability of Italian ryegrass (IRG) production to climate change, focusing on climate factors, a data analysis was performed to obtain an optimal model by applying various regression models. A correlation analysis was conducted using the eight climate factors listed in Table 1 as independent variables, while the dry matter yield (primary) data—the Italian ryegrass production data—served as the dependent variable. A production prediction model was then developed through a multiple regression analysis, the Lasso model, and the Ridge model. Subsequently, the production was compared and analyzed using these models.

Three regression models were estimated within the multivariate regression framework using a stepwise selection method in SPSS. All three models from the stepwise selection process were found to be significant. However, the model of the result of the third step was chosen because it had the highest value of adjusted $R^2$.

Table 3 summarizes the results of the multivariate regression model with the stepwise selection method using SPSS. As shown in Table 3a, as a result of the multiple regression analysis, the value of the determinant coefficient, $R^2$, was 0.446, which was considered to

be somewhat insufficient for prediction accuracy. Table 3b shows the selected variables with only the most influence as a result of the stepwise selection method. The results derived that three variables, MINTJAN, PREOCT, and GDDFJTA, were the key impact factors related to IRG production among the climate variables. Table 3c shows the results of summarizing the coefficients and intercepts of the independent variables adopted in the final selection model.

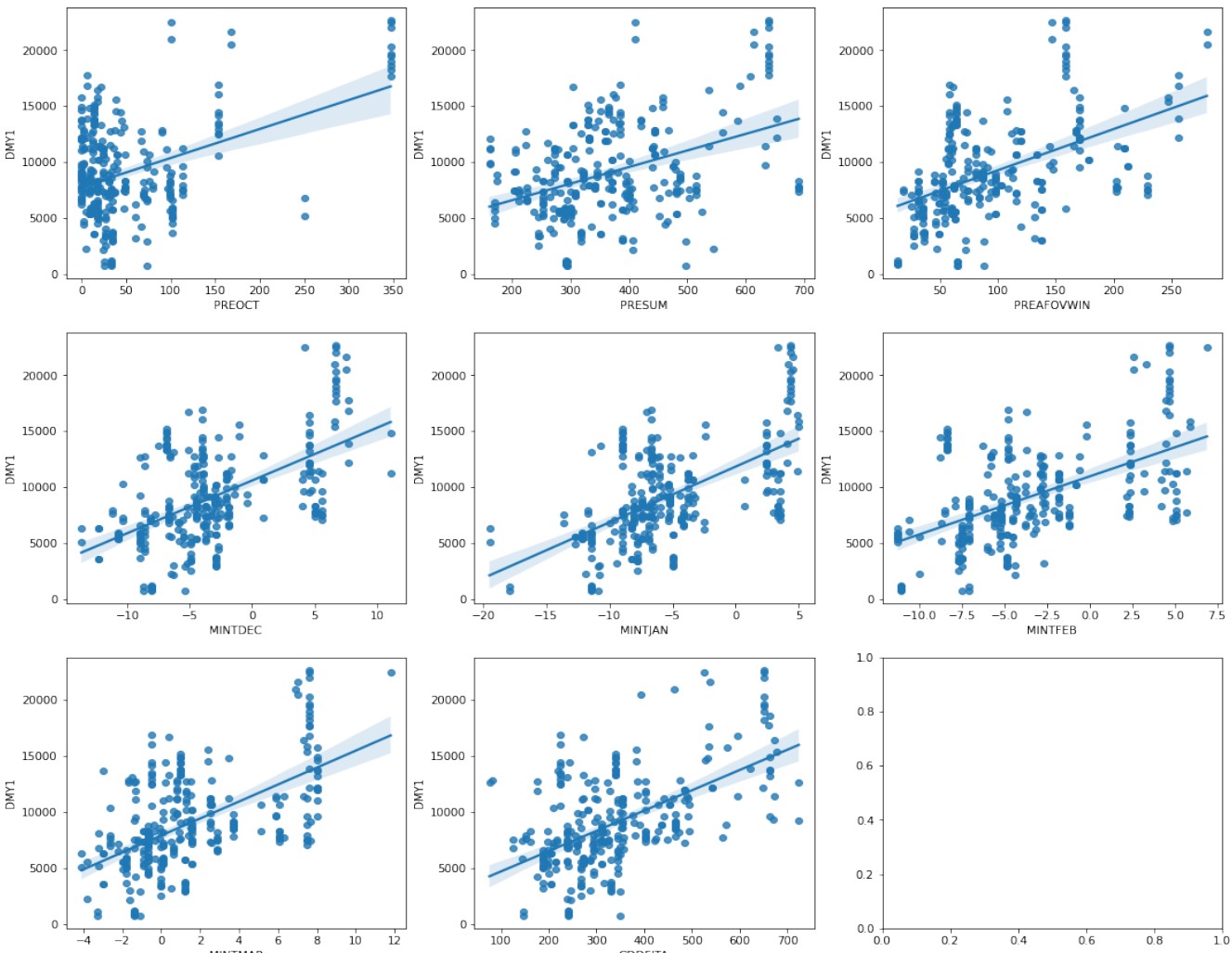

**Figure 5.** Graphs of correlation analysis results with IRG dataset.

As a result of the multiple regression analysis using SPSS, the value of the adjusted $R^2$ was 0.446, which was somewhat inadequate for determination. In addition, a problem of multicollinearity was identified due to the high correlation among the independent climate factors. Existing linear regression models often suffer from overfitting by closely analyzing the relationship between feature and label values in excessive detail. As a result, their ability to accurately predict new data is diminished due to poor generalization.

To address this issue, the Lasso model was employed as a regression technique. In linear regression, the essence lies in discovering suitable weights and biases. The Lasso model achieved this by minimizing the mean squared error (MSE) while introducing supplementary constraints, thereby simultaneously minimizing the sum of absolute weight values.

**Table 3.** Results of IRG multivariate regression analysis (stepwise selection method) (a): coefficient of determination ($R^2$) of the final selection model. (b): outcome variable of final selection model. (c): coefficient of final selection model.

| (a) | | | | | |
|---|---|---|---|---|---|
| Model Summary | | | | | |
| Model | R | R Squared | Modified R Squared | Standard Error of the Estimate | Durbin–Watson |
| 3 | 0.672 | 0.451 | 0.446 | 3197.9769 | 0.744 |

| (b) | | | | | |
|---|---|---|---|---|---|
| ANOVA [a] | | | | | |
| Model | | Sum of Squares | Degree of Freedom | Mean Square | F | Significance Probability |
| 3 | Regression | 2,498,119,243.996 | 3 | 832,706,414.665 | 81.422 | 0.000 |
| | Residual | 3,037,435,744.456 | 297 | 10,227,056.379 | | |
| | Total | 5,535,554,988.452 | 300 | | | |

| (c) | | | | | | | |
|---|---|---|---|---|---|---|---|
| Coefficient [a] | | | | | | | |
| Model | | Non-Standardized Coefficient | | Standardized Coefficient | t | Significance Probability | Collinearity Statistic | |
| | | B | Standard Error | Beta | | | Tolerance | VIF |
| 3 | (Constant) | 6167.375 | 1046.588 | | 5.893 | 0.000 | | |
| | MINTJAN | 231.498 | 58.153 | 0.273 | 3.981 | 0.000 | 0.392 | 2.551 |
| | PREOCT | 19.046 | 2.746 | 0.310 | 6.935 | 0.000 | 0.925 | 1.081 |
| | GDDFJTA | 9.382 | 2.166 | 0.291 | 4.332 | 0.000 | 0.411 | 2.435 |

[a] Dependent variable: DMY1; Predictor: (Constant), MINTJAN, PREOCT, GDDFJTA.

The Lasso model addressed the issue of multicollinearity during the modeling process, resulting in the creation of a final model. Therefore, the issue of the multiple regression model could be solved. In addition, the analysis using the Lasso model resulted in an improved determination coefficient ($R^2$) value of 0.6, which demonstrated a greater enhancement in the explanatory power compared to that of the existing multiple linear regression model. Specifically, the application of the Lasso regression model using the climate factors selected through the stepwise selection method resulted in an $R^2$ value of 0.629. As a result, the issue of multicollinearity was resolved, leading to an enhancement in the explanatory power of the prediction model.

This improvement enabled the derivation of an IRG production prediction model comprising key climate factors such as the PREOCT (precipitation in October), MINTJAN (minimum temperature in January), and GDDFJTA (growing degree days from January to April), which exerted a significant influence. To find the optimal IRG production prediction model, experiments were performed after reconstructing various datasets. Experiments were conducted for cases where the numbers of x_features were seven, four, and three, respectively, by varying the composition of the independent variables.

Table 4 shows the comparison results of several models, summarizing the experimental results. When applying the Lasso model with x_features = ['PREOCT', 'PRESUM', 'PREAFOVWIN', 'MINTDEC', 'MINTJAN', 'MINTFEB', 'MINTMAR', 'GDDFJTA'], which included all seven selected independent variables in the IRG dataset with N = 304, the value of the adjusted $R^2$ was confirmed to be 0.601. In the case of the Lasso model with x_features = ['PREOCT', 'MINTJAN', 'MINTFEB', 'GDDFJTA'], which selected four attributes with high importance from the independent variables in the IRG dataset with N = 304, the value of the adjusted $R^2$ was confirmed to be 0.627. When applying the Lasso model with x_features = ['PREOCT', 'MINTJAN', 'GDDFJTA'], which represented a multivariate regression model using a stepwise selection method among independent variables in the IRG dataset with N = 304, the value of the adjusted $R^2$ was 0.629, which was confirmed to be the highest determination coefficient. As a result, the climate factors, most influencing to the production of IRG, were the precipitation in October (PREOCT), minimum temperature in January (MINTJAN), and GDD from January to April (GDDFJTA). Considering Lasso's ability to enhance prediction accuracy by forcing regression coefficients of irrelevant variables to become zero, along with its capacity to yield a more interpretable model, it could be inferred that appropriate variables were selected.

**Table 4.** Comparison of IRG production prediction models.

| | Variables | Coefficients | Intercept | $R^2$ |
|---|---|---|---|---|
| Multivariate regression (stepwise) (N = 304, X = 7) | DMY1~ PREOCT MINTJAN GDDFJTA | 19.046 231.498 9.382 | 6167.375 | 0.34 |
| Lasso regression (N = 304, X = 7) | DMY1~ PREOCT PRESUM PREAFOVWIN MINTDEC MINTJAN MINTFEB MINTMAR GDDFJTA | 18.87 0.12 9.459 −122.77 90.58 101.65 227.68 7.51 | 4765.62 | 0.6019 |
| Lasso regression (N = 304, X = 4) | DMY1~ PREOCT MINTJAN MINTFEB GDDFJTA | 18.4995 120.31 140.11 9.2 | 6208.17 | 0.627 |
| Lasso regression (N = 304, X = 3) | DMY1~ PREOCT MINTJAN GDDFJTA | 18.778 239.335 8.906 | 6441.169 | 0.629 |

*3.2. Grass Production Prediction Model*

Unlike winter forage crops, grass is cultivated year-round. Its nature of being used for more than two years after initial growth means that any potential impact from climate factors is not significant. It can readily recover from such effects. Therefore, selecting climate factors for grass is more challenging than for winter forage crops. Fifteen climatic indicators that could affect the dry matter yield of grass were selected and tested primarily. However, it was judged that production prediction using the main indicators was desirable to study changes in the suitable cultivation area. Therefore, a production prediction model for grass was developed by selecting five climatic factors that were judged to have the most influence on grass production. Table 5 summarizes the descriptions of the five selected climate factors and the descriptive statistics of each variable. Among the climate factors, the drought indicator was obtained from the collected daily data, not the data crawled by the Korea Meteorological Administration. The DDAYS was obtained by counting the number of days without rain for more than 10 days. A correlation analysis was conducted based on the empirical knowledge that grass is influenced by precipitation over a longer duration compared to winter forage crops. The analysis involved the following variables: drought days (DDAYS), the sum of precipitation days in August (PREDAUG), the growing degree days from January to December (GDDTOTAL), the maximum temperature in July (MAXJUL), the maximum temperature in August (MAXAUG), and the total dry matter yield (DMYT). The results of the correlation analysis between these five climate indicators and the DMYT are presented in Table 6. The dependent variable, DMYT, was confirmed to satisfy normality.

The results of the correlation analysis between the total dry matter yield and five climate factors showed that the DDAYS, PREDAUG, MAXJUL, MAXAUG, and GDDTOTAL had slightly higher correlation coefficients. It could be seen that all independent variables had significant ($p < 0.01$) correlations with the dependent variable, DMYT, the total dry matter yield. Figure 6 shows a graph of the correlation analysis results for each variable.

**Table 5.** Variables and descriptive statistics of forage grass dataset.

| Category | Variable | Meaning | Mean | S.E. |
|---|---|---|---|---|
| Yield | DMYT | Total Dry Matter Yield | 15,854.71 | 4023.565 |
| Precipitation | DDAYS | Drought Days | 1.68 | 3.209 |
| | PREDAUG | Sum of Precipitation Days in August | 14.07 | 4.130 |
| Temperature | MAXJUL | Maximum Temperature in July | 34.0262 | 1.70214 |
| | MAXAUG | Maximum Temperature in August | 34.9328 | 1.71471 |
| GDD | GDDTOTAL | GDD From January to December | 3789.4775 | 493.55510 |

**Table 6.** Results of correlation analysis between grass production and 5 climate indicators.

| | | DDAYS | PREDAUG | MAXJUL | MAXAUG | GDDTOTAL |
|---|---|---|---|---|---|---|
| Pearson | DMY1 | 0.444 | −0.381 | 0.181 | 0.055 | 0.444 |

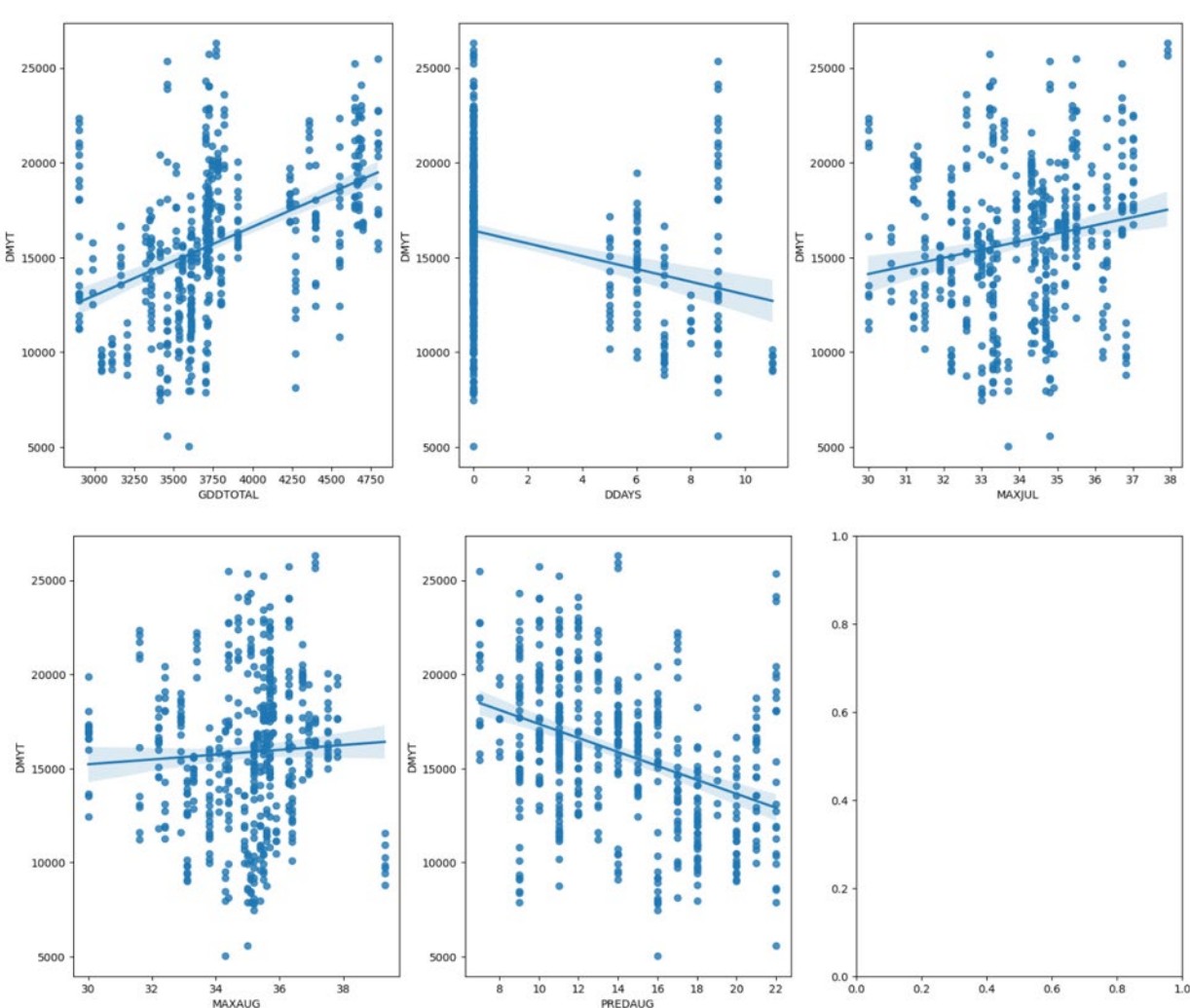

**Figure 6.** Graphs of correlation analysis results with the forage grass dataset.

To predict production, a production prediction model was derived using multiple regression analysis and the Lasso model. These two models were then compared with each other.

Two regression models were generated in the multivariate regression model with the stepwise selection method using SPSS. Both models were found to be significant. The second model was selected because its adjusted $R^2$ value was higher. A multiple regression

analysis was performed using SPSS, yielding a determination coefficient ($R^2$) of 0.23, slightly lower than that of the IRG. Table 7 summarizes the results of the multivariate regression model with the stepwise selection method using SPSS. As shown in Table 7a, as a result of the multiple regression analysis, the value of the determinant coefficient ($R^2$) was 0.230, which was considered to be somewhat insufficient for determination. Table 7b shows the selected variables with only the most influential as a result of the stepwise selection method. The results of the final selection model in the stepwise selection method showed that the PREDAUG and GDDTOTAL affected grass production in Table 7b. Table 7c shows the coefficients and intercepts of the final model. Grass was harvested year-round through the establishment of grasslands, showing a distinct contrast from the winter forage crops.

**Table 7.** Results of the grass multivariate regression analysis (stepwise selection method) (a): determination coefficient ($R^2$) of the final selection model. (b): result valuables of the final selection model. (c): coefficients of the final selection model.

| (a) | | | | |
|---|---|---|---|---|
| Model | R | R Squared | Adjusted R Squared | Standard Error of the Estimate |
| 2 | 0.482 | 0.233 | 0.230 | 3531.551 |

| (b) | | | | | |
|---|---|---|---|---|---|
| ANOVA [a] | | | | | |
| Model | Sum of Squares | Degree of Freedom | Mean Square | F | Significance Probability |
| Regression | 1,879,834,560.268 | 2 | 939,917,280.134 | 75.363 | 0.000 |
| 2  Residual | 6,198,512,188.682 | 497 | 12,471,855.510 | | |
| Total | 8,078,346,748.950 | 499 | | | |

| (c) | | | | | |
|---|---|---|---|---|---|
| Model | Non-Standardized Coefficient | | Standardized Coefficient | t | Significance Probability |
| | B | Standard Error | Beta | | |
| (Constant) | 8380.649 | 1779.579 | | 4.709 | 0.000 |
| 2  PREDAUG | 2.757 | 0.367 | 0.338 | 7.521 | 0.000 |
| GDDTOTAL | −211.288 | 43.808 | −0.217 | −4.823 | 0.000 |

[a] Dependent variable: DMYT; Predictor: (constant), PREDAUG, GDDTOTAL.

Experiments were planned, applying the reorganized multivariate datasets, to figure out the optimal prediction model for forage production. Experiments were performed, varying the sample size (N) and the configuration of the independent variables, as shown in Table 8.

Table 8 summarizes the experimental outcomes and presents a comparison of several models. When applying the Lasso model with x_features = ['DDAYS', 'PREDAUG', 'MAXJUL', 'MAXAUG', 'GDDTOTAL'], which includes all five selected independent variables in the grass dataset with N = 500, the value of the adjusted $R^2$ improved to 0.4799, demonstrating an enhanced explanatory power compared to the multiple regression model. When applying quartile refinement to data within the 25% to 75% range and experimenting with data sizes of N = 400 and N = 300, respectively, the values of the adjusted $R^2$ did not show significant improvement. As shown in Table 8, it was discovered that, in the case with a data size of N = 400, the value of the adjusted $R^2$ was 0.367, and in the case with a data size of N = 300, the value of the adjusted $R^2$ was 0.288. This result was interpreted as not requiring additional filtering because the grass data had normality. In the case of the Lasso model, incorporating all 15 climate factors that were initially selected for the first time, the adjusted $R^2$ value was 0.48, indicating a slight improvement. This outcome can be interpreted as an enhancement in the determination coefficient, attributed to increases in the independent variables. If the $R^2$ values were comparable, it was desirable to develop a productivity prediction model using the minimum number of indicators. If there were numerous indicators, it became possible to evaluate various factors. However, managing the fundamental data became challenging. When the value of the determination coefficient ($R^2$) was satisfied, selecting a smaller number of independent variables could simplify their future use in the analysis of suitable cultivation area changes. It was found that the

climate factors that most affected the production of grass were the PREDAUG, GDDTOTAL, MAXJUL, MAXAUG, and DDAYS. Therefore, we selected the Lasso model including five climate factors as an optimal model. The second row of Table 8 specifies the coefficients and intercepts for each variable of the optimal model.

**Table 8.** Comparison of results from analysis of the grass production prediction model.

|  | Variables | Coefficients | Intercept | $R^2$ |
|---|---|---|---|---|
| Multivariate Regression (Stepwise) (N = 500, X = 5) | DMY1~ PREDAUG GDDTOTAL | −211.288 2.757 | 8380.649 | 0.23 |
| Lasso Regression (N = 500, X = 5) | DMY1~ DDAYS PREDAUG MAXJUL MAXAUG GDDTOTAL | 53.6 −205.95 −129.12 −7.35 3.244 | 11,016.29 | 0.4799 |
| Lasso regression (N = 400, X = 5) | DMY1~ DDAYS PREDAUG MAXJUL MAXAUG GDDTOTAL | 8.305 −133.099 −97.403 12.386 2.096 | 12,613.215 | 0.367 |
| Lasso regression (N = 300, X = 5) | DMY1~ DDAYS PREDAUG MAXJUL MAXAUG GDDTOTAL | 21.278 −25.871 170.076 36.197 1.1505 | 5119.838 | 0.288 |
| Lasso regression (N = 500, X = 8) | DMY1~ DDAYS PREDAUG MAXJUL MAXAUG GDDTOTAL MINTFEB MINTMAR MTMAR | 73.392 −235.871 −11.940 250.936 −1.031 314.83 −170.95 445.2049 | 13,656.397 | 0.434 |
| Lasso regression (N = 500, X = 15) | DMY1~ DDAYS PREDAUG MAXJUL MAXAUG GDDTOTAL GDDFJTA PREDNOV MINTJAN MINTFEB MINTMAR MAXTNOV MAXTMAR MTJAN MTFEB MTMAR | 64.391 −266.45 −33.03 378.06 −0.7308 −3.647 94.77 11.742 282.09 −224.659 −220.732 −171.907 −224.456 326.645 915.632 | 13,581.535 | 0.480 |

## 4. Electromagnetic Climate Map for Suitable Cultivation Areas for Forage Crops

The production prediction model developed in Section 3, based on the climate impacts of grasses and IRG, could explain the vulnerability of forage crops to climate effects. In addition, it could facilitate the prediction of changes in the cultivation areas of grasses and IRG in response to various climate change scenarios. However, the electromagnetic climate map in this study only dealt with the electromagnetic climate map of the movement of the suitable cultivation area, which was built by limiting items and criteria, such as the soil and variety, for the climate effect.

The reference value for the cultivation area of IRG, a winter forage crop, was established based on the average minimum temperature in January. The best suitable areas were defined as areas with temperatures of −5 °C or above. The suitable areas were defined as areas with temperatures of −9 °C or above and potential areas were defined as areas with temperatures of −12 °C or above. The five-year average minimum temperature in January, spanning from 2017 to 2021, established the baseline. Additionally, the analysis value represented the average minimum temperature in January as of 2022. Figure 7 shows an electromagnetic climate map for the IRG cultivation area. The ratio of suitable cultivation area for IRG is shown in Table 9. For the analysis value, an increase was observed in areas with low production, while a decrease occurred in the optimal area due to temperatures lower than usual. In general, the proportion of the area corresponding to the analysis value for suitable or better conditions decreased compared to the reference value, whereas the proportion for the possible or low-production area increased.

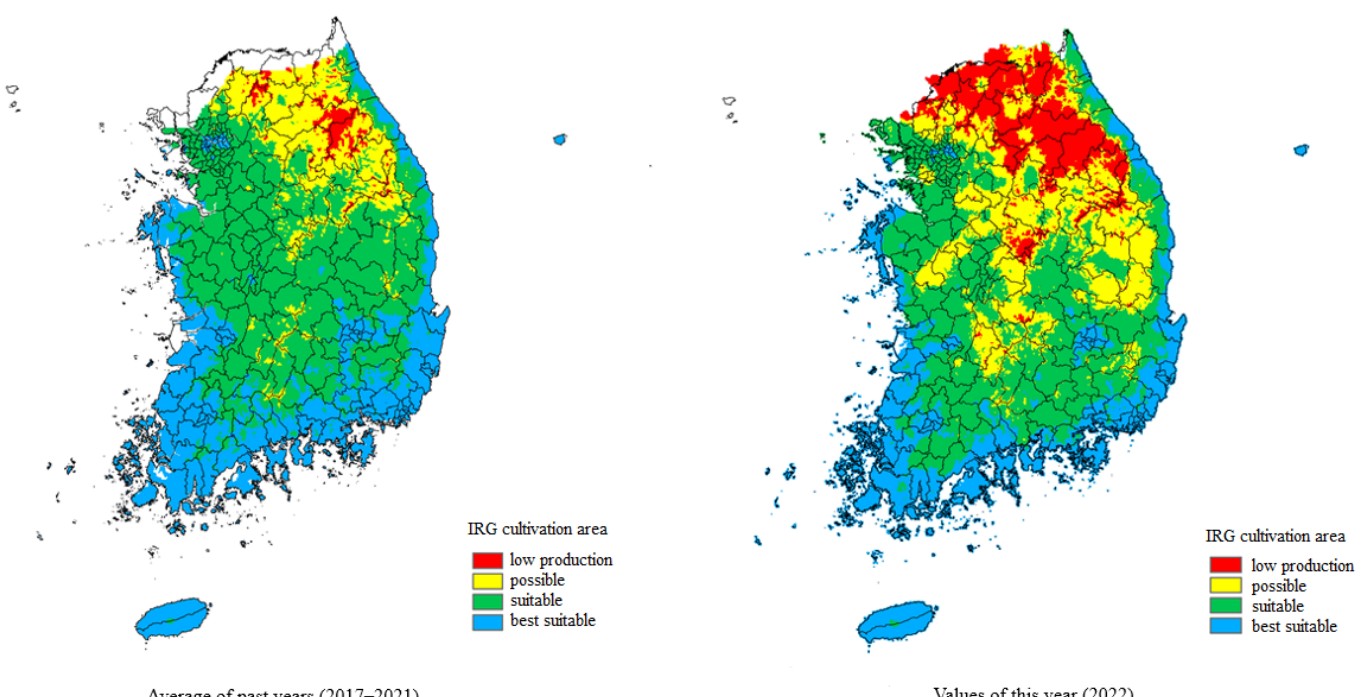

**Figure 7.** Electromagnetic climate map of suitable cultivation areas for IRG.

**Table 9.** Ratio of suitable cultivation areas for IRG.

| Category (AVGMINTJAN) | Best Suitable (≥−5 °C) | Suitable (−5 °C~−9 °C) | Possible (−9 °C~−12 °C) | Low Production (<−12 °C) |
|---|---|---|---|---|
| Result of Year (2022) | 27% | 39% | 22% | 12% |
| Result of Past 5 Years (2017~2021) | 32% | 51% | 15% | 2% |

The criterion for the suitable grass cultivation area was defined based on the average maximum temperature in August. The optimal area was set for areas with temperatures

below 25 °C. The suitable area was set for areas with temperatures of 26 to 28 °C. The possible area was set for areas with temperatures of 29 to 31 °C and the low production area was set for areas with temperatures over 32 °C. The reference value used to create the electromagnetic climate map was established as the five-year average maximum temperature in August from 2017 to 2021. Table 10 shows the ratio of suitable cultivation areas for forage grass.

**Table 10.** Ratio of suitable cultivation areas for forage grass.

| Category (AVGMAXTAUG) | Best Suitable (≤25 °C) | Suitable (26 °C~28 °C) | Possible (29 °C~31 °C) | Low Production (≥32 °C) |
|---|---|---|---|---|
| Result of year (2022) | 16% | 64% | 20% | |
| Result of past 5 years (2017~2021) | 8% | 29% | 63% | |

The analysis value, on the other hand, was determined as the average maximum temperature in August for the year 2022. As shown in Figure 8, the analysis of suitable grass cultivation areas revealed an increase in the proportion of suitable or optimal areas compared to the reference value. This increase was attributed to lower summer temperatures compared to typical years. It is anticipated that accurate predictions of changes in cultivation area movement can be achieved by gathering additional data and developing an electromagnetic climate map in the coming years.

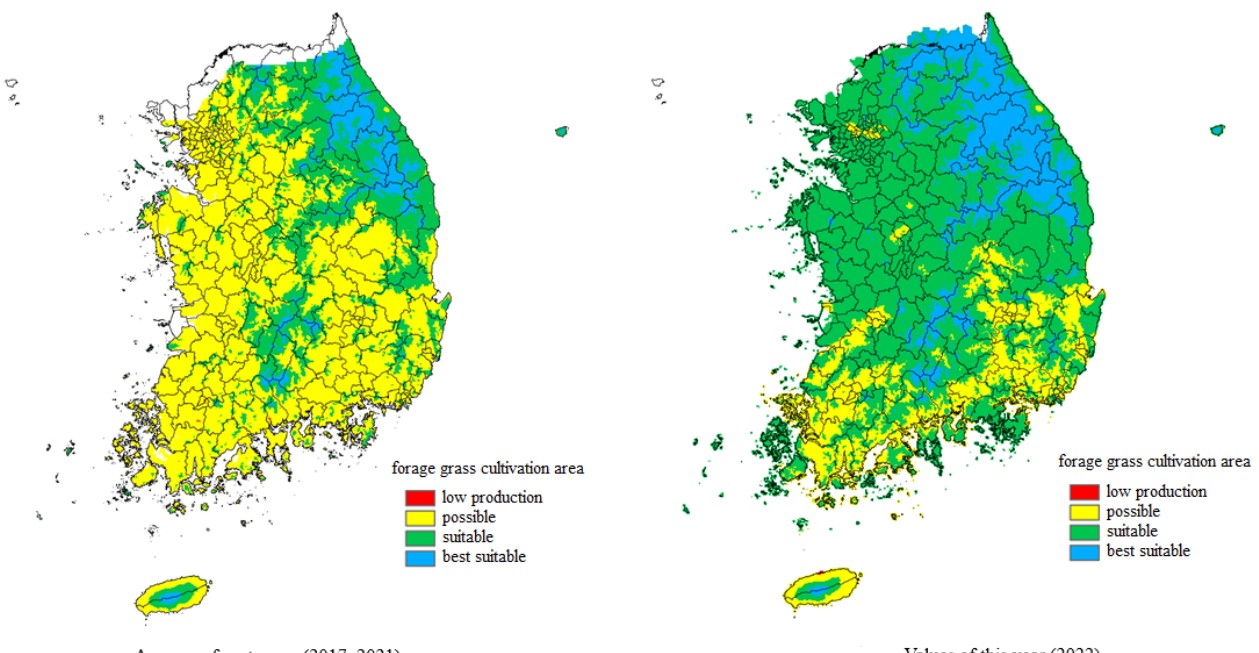

Average of past years (2017–2021)

Values of this year (2022)

**Figure 8.** Electromagnetic climate map of suitable cultivation areas for forage grass.

Figure 9 shows the electromagnetic map used for predicting forage crop productivity. The map reveals that the maximum yield, reaching 16,151 kg/ha, is predominantly concentrated in the Jeju region and certain coastal areas. Conversely, the minimum yield, at 426.9 kg/ha, is primarily observed in the alpine regions in Gangwon-do as well as mid-northern and high-altitude areas.

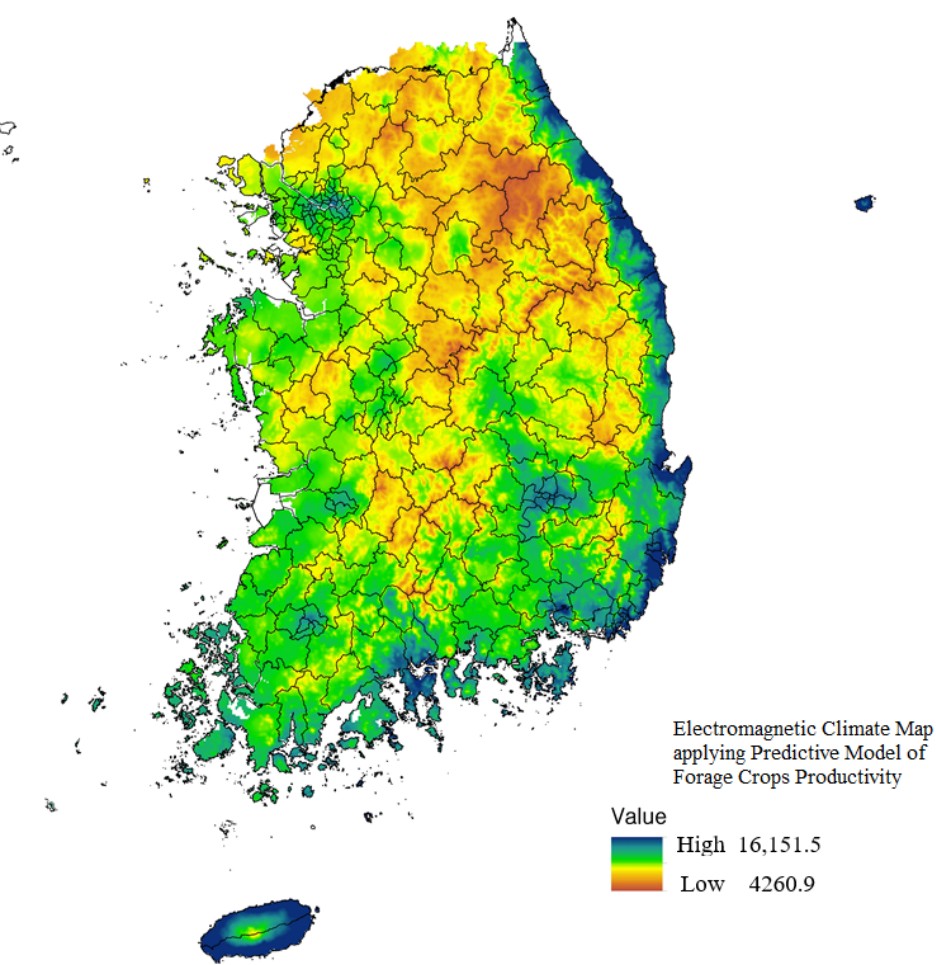

**Figure 9.** Electromagnetic climate map applying the predictive model of forage crop productivity in Republic of Korea (2022).

## 5. Conclusions

In this paper, we analyzed forage crop productivity and changes in suitable cultivation areas based on climate change in Republic of Korea. The dry matter data of IRG and forage with labels of region and year were collected through literature reviews and experimental reports from the Korea Institute of Animal Science and Technology. In addition, we developed a climate data crawler, through which climate data were crawled from the website of the Korea Meteorological Administration, that gathered climate data according to the region and year labels. Subsequently, the collected data were preprocessed to establish a foundational dataset database. By applying various regression models, predictive models that could predict future production were compared and analyzed, and major climate factors affecting production were determined. As a result of the production prediction model, the Lasso model with the highest determination coefficient, $R^2$, was selected to derive climate factors that greatly affected each forage crop. Precipitation in October (PREOCT), the minimum temperature in January (MINTJAN), and the growing degree days from January to April (GDDFJTA) were identified as key climate factors with the greatest impact on Italian ryegrass production. The climate factors that had the greatest impact on grass production were the drought days (DDAYS), the sum of precipitation days in August (PREDAUG), the growing degree days from January to December (GDDTO-TAL), the maximum temperature in July (MAXJUL), and the maximum temperature in August (MAXAUG).

Based on the results of the analyses, electromagnetic climate maps were constructed for suitable cultivation areas and production prediction. In Republic of Korea, the ratio of optimal areas was high in the southern region and the ratio of suitable areas was high in the central region. In the case of the mid-northern mountainous area, the ratio of possible and low-production areas was relatively high. Therefore, mid-northern and mountainous regions are likely to be advantageous in terms of productivity to use domestically developed IRGs with high cold resistance. Grasses are categorized as C3 crops. Their economic lifespan is often shortened due to high temperatures in Korea. Given that the optimal temperature range is between 20 $^\circ$C and 25 $^\circ$C, the optimal and suitable areas are primarily concentrated in the mid-mountainous area of the mid-northern region and the mid-mountainous area of Jeju Island. In the case of the central and southern regions, there are many possible and low-production areas. Thus, it seems necessary to introduce high-temperature-resistant varieties or C4 grasses that can replace northern grasses in the future.

Our contribution is that we have built a database related to feed crops, which exists rarely. In addition, the research results can be used for livestock farms to establish policies to prepare against climate change. The limitation of our study is that it excluded various factors such as soil, fertilizer, breed, and so on, which affect the production of feed crops, and that it was carried out to only focus on analyzing climate data with impacts on feed crop production. The value of $R^2$, the determination coefficient, was considered to be not much higher due to these limitations.

In future work, we will collect feed crop data and climate data to predict suitable cultivation areas, so as to build an electronic climate map, and publish it for farming guidance for livestock farms.

**Author Contributions:** Conceptualization, S.H., J.-S.J. and M.S.; methodology, S.H. and J.-S.J.; investigation, J.K. and B.K.; validation, B.K. and S.H.; writing—original draft, J.-S.J. and M.S.; writing—review and editing, J.K. and M.S.; funding acquisition, M.S. All authors have read and agreed to the published version of the manuscript.

**Funding:** This research was funded by the Cooperative Research Program for Agriculture Science and Technology Development (No. PJ015079032023), Rural Development Administration, Republic of Korea.

**Institutional Review Board Statement:** Not applicable.

**Informed Consent Statement:** Not applicable.

**Data Availability Statement:** The data presented in this study are available on request from the corresponding author.

**Conflicts of Interest:** The authors declare no conflict of interest.

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
