# Peer review of "A Study on Analyses of the Production Data of Feed Crops and Vulnerability to Climate Impacts According to Climate Change in Republic of Korea"

_applsci, doi:10.3390/app132011603_

Round 1

Reviewer 1 Report

The manuscript discusses the impact of climate change on the  forage crop production. A production prediction model  was proposed based on climate indicators based to predict the future yields.  The following points are observed. 

1. The abstract is too lengthy. Make it short and Crisp. The rest of the portion can be placed in Introduction section. 

2. Keywords are not properly defined. Please read the journal instructions on using appropriate keywords. Avoid using full sentences as keywords like "Predictive Model of Forage Crops Productivity", "Suitable Cultivation Area" etc. These cannot serve as keywords. 

3. Resolution of Figure 1 is very poor. It is suggested to use high resolution images atleast of 300 to 600 dpi . 

4. Figure 2 must be redrawn. It is not visible properly. 

5. Is there any standard research methodological approach being used or the authors have used there own methodology. Please clarify as the research methodology is not provided in much details.

6.  Literature Review is missing. Only one paper of 2020 is present in the references. It is suggested to include a literature review section and include latest state-of-the-art on the topic.  

The manuscript discusses the impact of climate change on the  forage crop production. A production prediction model  was proposed based on climate indicators based to predict the future yields.  The following points are observed. 

1. The abstract is too lengthy. Make it short and Crisp. The rest of the portion can be placed in Introduction section. 

2. Keywords are not properly defined. Please read the journal instructions on using appropriate keywords. Avoid using full sentences as keywords like "Predictive Model of Forage Crops Productivity", "Suitable Cultivation Area" etc. These cannot serve as keywords. 

3. Resolution of Figure 1 is very poor. It is suggested to use high resolution images atleast of 300 to 600 dpi . 

4. Figure 2 must be redrawn. It is not visible properly. 

5. Is there any standard research methodological approach being used or the authors have used there own methodology. Please clarify as the research methodology is not provided in much details.

6.  Literature Review is missing. Only one paper of 2020 is present in the references. It is suggested to include a literature review section and include latest state-of-the-art on the topic.  

Author Response

Response to Reviewer 1 Comments

To the Reviewers and Editor:

We are very pleased to have received your valuable comments regarding our manuscript entitled “A Study on Analysis of Production Data of Feed Crops and Vulnerability to Climate Impacts According to Climate Change in South Korea”.

In the revision step, we have revised our manuscript according to the editor’s and reviewers’ comments, and our revisions are provided below.

Point 1: The abstract is too lengthy. Make it short and Crisp. The rest of the portion can be placed in Introduction section

Response 1: The abstract has been revised. It was reorganized to be short and concise, and the remaining parts were revised in the introduction section.

Point 2: Keywords are not properly defined. Please read the journal instructions on using appropriate keywords. Avoid using full sentences as keywords like "Predictive Model of Forage Crops Productivity", "Suitable Cultivation Area" etc. These cannot serve as keywords. 

Response 2: The keyword has been modified. It was reorganized into words rather than full sentences. 

Point 3: Resolution of Figure 1 is very poor. It is suggested to use high resolution images at least of 300 to 600 dpi . 

Response 3: Figure 1 has been replaced with a high-resolution image.

Point 4: Figure 2 must be redrawn. It is not visible properly. 

Response 4: Figure 2 is the UI of the climate data crawler developed using Python to collect climate data required for this study. Because images of figure 2 are the screen capture images while data crawlling, they cannot be redrawn. Figure 2 shows to present the process of climate data collection. The UI is simple, but it is functionally efficient as it was developed using Python.

Point 5: Is there any standard research methodological approach being used or the authors have used there own methodology. Please clarify as the research methodology is not provided in much details.

Response 5:  Our research methodology is added in Introduction section. The goal of our study is to predict forage crop production and build an electronic climate map according to climate change. 1) Collection of pasture/fodder crop production data - 2) Collection of climate data according to the year and region of the collected production data - 3) Collection of collected production data Selection of the most influential climate elements from climate data - 4) Creation of various regression models and analysis of results - 5) Establishment of a production prediction model with climate factors that affect each feed crop - 6) Construction of an electronic climate map accordingly. This can serve as basis for the movement of suitable cultivation areas for feed crops due to climate change on the Korean Peninsula, and is planned to be used for farming guidance for livestock farmers in South Korea.

Point 6: Literature Review is missing. Only one paper of 2020 is present in the references. It is suggested to include a literature review section and include latest state-of-the-art on the topic

Response 6: Some of latest literature were added to the references and cited in the paper. In South Korea, many production forecasts have been conducted for general crops such as rice, barley, and cabbage, but there has been little research on forecasting feed crop production for livestock farms. Only recently has South Korea been conducting research and establishing policies to support livestock and livestock farms due to abnormal climate, and the Korea Institute of Animal Science is also conducting research on the vulnerability of feed crops due to climate change, and thus the latest research on forecast models for feed crop production in South Korea. There are not enough literatures for reference. So we have figured it out. Our contribution is to apply big data analysis technology to build a production predictive model for forage crops and improve accuracy by applying lasso model and ridge model.

Regarding the quality of English, we had our paper edited by an expert, and a certificate was attached.

We believe that the topic of our paper is appropriate for the special issue "Recent Advances in Precision Farming and Digital Agriculture".

We greatly appreciate your helpful comments, which we feel have improved our manuscript.

Reviewer 2 Report

The entitled "A Study on Analysis of Production Data of Feed Crops and vulnerability to Climate Impacts According to Climate Change in South Korea" presented a research of agricultural production on Korean Peninsula under the warming climate. The whole manuscript was more like a technical report rather than a scientific research paper. From the assessment of data and method, there were nothing new. The data was from the public website, and the methods were all commonly applied regression method and deep learning ones. I would recommend the rejection for this manuscript (see comments below).

For specific comments:

1. what is ‘deep learning regression’? clarify. This is the first time I see this kind of expression.

2. line 116 to 119. Repeated references compared with above. The manuscript should be well re-constructed. The current one is in a mess.

3. line 86. ‘and’ is in wrong format.

4. figure 5. The scatter points shown indicated that the relationships between the independent and dependent variables were not strong.

5. figure 6. This is very weird. The same x values for many different y values. There must be something wrong for this.

6. reference 1-6 were all from the same person ‘kim’. This is not acceptable.

7. For machine learning and deep learning part. I would suggest some references of applications using UAV for this:

1) Machine Learning-Based Approaches for Predicting SPAD Values of Maize Using Multi-Spectral Images[J]. Remote Sensing, 2022, 14(6): 1337.

2) "Identifying brown bear habitat by a combined GIS and machine learning method." Ecological Modelling 135.2-3 (2000): 291-300.

minor revision

Author Response

Response to Reviewer 2 Comments

To the Reviewers and Editor:

We are very pleased to have received your valuable comments regarding our manuscript entitled “A Study on Analysis of Production Data of Feed Crops and Vulnerability to Climate Impacts According to Climate Change in South Korea”.

In the revision step, we have revised our manuscript according to the editor’s and reviewers’ comments, and our revisions are provided below.

Point 1: What is ‘deep learning regression’? clarify. This is the first time I see this kind of expression.

Response 1: Deep learning regression refers to regression analysis, a branch of machine learning. The Lasso model applied in our paper, Ridge model, and Elastic model are called machine learning(deep learning) regression regression because they utilize the Scykit-Learn package, which including libraries useful in Python's machine learning analysis. The reason for applying a machine learning regression model is that climate factors are highly correlated in the forage crop production prediction model, so multicollinearity problems occur in statistical regression models. In cases where there is inevitably a high correlation between independent variables, Ridge regression or Lasso regression are known to be more appropriate approaches, and regularization techniques prevent overfitting.

Point 2: line 116 to 119. Repeated references compared with above. The manuscript should be well re-constructed. The current one is in a mess.

Response 2: The lines 116-119(->123-130) were revised.

Point 3: line 86. ‘and’ is in wrong format.

Response 3: The line 86 was revised.

Point 4: figure 5. The scatter points shown indicated that the relationships between the independent and dependent variables were not strong. 

Response 4:  As shown in figure 5, the correlation between forage crop production data and climate factors were not strong. This is because various factors, such as soil, variety, and fertilizer, as well as climate, have affected the growth of crops. In this study, we excluded factors other than climate, and attempted to analyze only climate factors and the influence of climate change.  

Point 5: figure 6. This is very weird. The same x values for many different y values. There must be something wrong for this.

Response 5: Figure 6 shows a graph of results of correlation analysis between these five climate indicators and DMYT. The analysis involved the following variables: Drought Days (DDAYS), Sum of Precipitation Days in August (PREDAUG), Growing Degree Days from January to December (GDDTOTAL), Maximum Temperature in July (MAXJUL), Maximum Temperature in August (MAXAUG), and Total Dry Matter Yield (DMYT). And data set as shown in Figure 4, Total Dry Matter Yield is different depending on the region, but climate data such as Drought Days (DDAYS), Sum of Precipitation Days in August (PREDAUG), Growing Degree Days from January to December (GDDTOTAL), Maximum Temperature in July (MAXJUL), Maximum Temperature in August (MAXAUG) in the same year are the same, so the graph was derived as shown in Figure 6.

Point 6: reference 1-6 were all from the same person ‘kim’. This is not acceptable.  

Response 6: The reference 1-6 were revised.

Point 7: For machine learning and deep learning part. I would suggest some references of applications using UAV for this:1) Machine Learning-Based Approaches for Predicting SPAD Values of Maize Using Multi-Spectral Images[J]. Remote Sensing, 2022, 14(6): 1337. 2) "Identifying brown bear habitat by a combined GIS and machine learning method." Ecological Modelling 135.2-3 (2000): 291-300

Response 7: Thanks for the recommending above papers. It seems valuable as an example of machine learning application. However, our data is based on pasture crop production data and related climate data, and the Lasso model was confirmed as the most appropriate among multiple regression models. Thank you for allowing me to refer to good literature.

Regarding data used in this paper, IRG data and forage data did not have taken from public sites. The IRG data and forage data have been collected through literature review and experimental reports from the Korea Institute of Animal Science and Technology. In addition, we developed a climate data crawler to collect climate data according to the region and year of the production data. We collected related climate data through executing the program and built a production prediction model to analyze the vulnerability of fodder crops to climate impacts. This study was conducted in collaboration with the Korea Institute of Animal Science and Technology.

Regarding the imprevement of English, we had our paper edited by an expert. A certificate was attached.

We believe that the topic of our paper is appropriate for the special issue "Recent Advances in Precision Farming and Digital Agriculture".

We greatly appreciate your helpful comments, which we feel have improved our manuscript.

Reviewer 3 Report

The authors worked on the analysis of production data of feed crops in terms of climate impacts. I have the following major comments:

It is quite difficult to get to know the main contributions of the paper. Therefore, the authors are suggested to add a separate paragraph before the section 2 and explain the main novelties of the paper in a detailed way.

The literature review is not complete. Therefore, to improve the paper, the authors must add the following missing references:

Analyzing climate change impacts on health, energy, water resources, and biodiversity sectors for effective climate change policy in South Korea. Sci Rep 11, 18512 (2021). https://doi.org/10.1038/s41598-021-97108-7

"Internet of Things (IoT) Assisted Context Aware Fertilizer Recommendation," in IEEE Access, vol. 10, pp. 129505-129519, 2022, doi: 10.1109/ACCESS.2022.3228160.

"Context Aware Evapotranspiration (ETs) for Saline Soils Reclamation," in IEEE Access, vol. 10, pp. 110050-110063, 2022, doi: 10.1109/ACCESS.2022.3206009.

The authors are suggested to explain all the used abbreviations before writing them, such as what is RCP 8.5?

The graphical quality of Fig. 1 is not up to the mark. Therefore, please provide a better-quality image.

The details of data collection is totally missing in section 2. Therefore, please provide all the missing information.

Why did not the authors use feature scaling techniques in Fig. 4?

What is the reason of using machine learning approaches instead of deep learning? Proper explanation is required to be added in the paper.

The comparison is missing in the paper. Therefore, please provide some comparison between the utilized method and the existing approaches to highlight the main novelty of the paper.

Readership appeal is moderate. Therefore, please provide more pictorial information.

The authors have not calculated any performance parameters of regression, such as MAE, etc. please add the missing performance parameters.

It is quite difficult to reproduce the study. Therefore, please provide the values of all parameters set during the simulations.

How did the authors get the figures 7 to 9? Please provide proper references.

Before the conclusion section, please add a new section and explain the limitations of the study there.

The conclusion section can be enriched with some future directions.

Minor editing of English language required

Author Response

Response to Reviewer 3 Comments

To the Reviewers and Editor:

We are very pleased to have received your valuable comments regarding our manuscript entitled “A Study on Analysis of Production Data of Feed Crops and Vulnerability to Climate Impacts According to Climate Change in South Korea”.

In the revision step, we have revised our manuscript according to the editor’s and reviewers’ comments, and our revisions are provided below.

Point 1:  It is quite difficult to get to know the main contributions of the paper. Therefore, the authors are suggested to add a separate paragraph before the section 2 and explain the main novelties of the paper in a detailed way.

Response 1: The abstract has been revised. It was reorganized, and the novelity and contribution of the paper were described.  

Point 2: The literature review is not complete. Therefore, to improve the paper, the authors must add the following missing references:

Response 2: The literature review were added and the references were revised.

Point 3: The authors are suggested to explain all the used abbreviations before writing them, such as what is RCP 8.5?

Response 3: We revised to explain all the used abbreviations before writing them.

Point 4: The graphical quality of Fig. 1 is not up to the mark. Therefore, please provide a better-quality image.

Response 4: Figure 1 has been redrawn.

Point 5: The details of data collection is totally missing in section 2. Therefore, please provide all the missing information.

Response 5:  The details of data collection were added in section 2. 

Point 6:  Why did not the authors use feature scaling techniques in Fig. 4?

Response 6:  Figure 4 showed a set of preprocessed data results. Scaling was performed with a Python libraries.

Point 7: What is the reason of using machine learning approaches instead of deep learning? Proper explanation is required to be added in the paper.

Response 7:  In the case of deep learning models, it is possible to predict the production of forage crops, but since it is not clear which attributes of climate factors affect the prediction model, so a machine learning approach was applied. In our research, we derived climate factors that affect feed crop production,and built an electronic climate map based on the results, that can be used to predict changes in suitable cultivation areas for each feed crop.

Point 8: The comparison is missing in the paper. Therefore, please provide some comparison between the utilized method and the existing approaches to highlight the main novelty of the paper.

Response 8: Although research has been performed on the production of many crops, including rice, wheat, barley, cabbage, and radish, not much research has been conducted on feed crops in Korea. Recently, in order to improve the productivity of livestock farms, research on the productivity of feed crops is necessary, so the National Institute of Animal Science conducted a survey on production data for each feed crop and studied experimental results. A major contribution of this study is that we collected production data through various sources, collected climate data according to the region and year of production data, and established a database related to feed crops. The existing approach was to experiment with soil, fertilizer, etc., but in this study, data was analyzed to figure out the relationship between forage crop production and climate impact, and an electronic climate map of changes in cultivation areas that could be provided to livestock farms was constructed. The comparison of a variety of models for various dataset which consists of different independent variables were performed because best model has been figured out as shown in Table 4 and Table 8.

Point 9: Readership appeal is moderate. Therefore, please provide more pictorial information.

Response 9: A diagram was added to illustrate the data collection and processing process.

Point 10:  The authors have not calculated any performance parameters of regression, such as MAE, etc. please add the missing performance parameters.

Response 10: The Coefficient of determination (R²) of each model was compared. The Lasso model, which was found to have the highest value of Coefficient of determination (R²) , was adopted to derive climate impact factors and construct an electronic climate map.

Point 11: It is quite difficult to reproduce the study. Therefore, please provide the values of all parameters set during the simulations.

Response 11: The source code of the Python program executed has not been included. Reproduction is possible only with the Python program source and dataset.  

Point 12: How did the authors get the figures 7 to 9? Please provide proper references.

Response 12: The electronic climate maps in Figure 7-9 were drawen by values of climatic factors derived from the production prediction model for suitable cultivation areas. Using GIS tools like arcGIS, it is possible to construct electronic climate map based on location coordinates and overlayered climate data.

Point 13: Before the conclusion section, please add a new section and explain the limitations of the study there.The conclusion section can be enriched with some future directions.

Response 13: Limitations of our study have been presented and conclusion section was revised.

Regarding the quality of English, we had our paper edited by an expert again.

We believe that the topic of our paper is appropriate for the special issue "Recent Advances in Precision Farming and Digital Agriculture".

We greatly appreciate your helpful comments, which we feel have improved our manuscript.

Round 2

Reviewer 1 Report

The suggested revisions have been incorporated.

Suggested revisions have been incorporated. 

Author Response

Dear Reviewer,

We are very pleased to have received your valuable comments regarding our manuscript entitled “A Study on Analysis of Production Data of Feed Crops and Vulnerability to Climate Impacts According to Climate Change in South Korea”.

We really appreciate your helpful comments, which we feel have improved our manuscript.

Reviewer 2 Report

The mauscript in low quality and the figures were poorly poltted. The only adoption of linear regression can not be acceptable. Therefore, the machine learning approaches should be added in section discussion.

The current status of manuscript is not acceptable.

 For machine learning and deep learning part. I would suggest some references of applications using UAV for this:1) Machine Learning-Based Approaches for Predicting SPAD Values of Maize Using Multi-Spectral Images[J]. Remote Sensing, 2022, 14(6): 1337. 2) "Identifying brown bear habitat by a combined GIS and machine learning method." Ecological Modelling 135.2-3 (2000): 291-300

These issues should be well revised.

well

Author Response

To the Reviewer 

We are pleased to receive your comments regarding our manuscript entitled “A Study on Analysis of Production Data of Feed Crops and Vulnerability to Climate Impacts According to Climate Change in South Korea”.

We have revised our manuscript again.

Thanks for the recommending reference papers. I referenced one of them in introduction as a related works. It seems meaningful applying machine learning approach so we discuss it as a future work. In this paper, we focused on to figure out effects of climate change to the pasture crop production data according to climate data, and predict suitable cultivation area for pasture crop. Lasso model was confirmed as the most appropriate among multiple regression models because independent variables, climate data, had multicollinearity.

Lasso Regression is a popular type of regularized linear regression that includes an L1 penalty. This has the effect of shrinking the coefficients for those input variables that do not contribute much to the prediction task. This penalty allows some coefficient values to go to the value of zero, allowing input variables to be effectively removed from the model, providing a type of automatic feature selection.

Thank you for allowing me to refer to good literature.

Reviewer 3 Report

The author’s response is not satisfactory because the manuscript is hardly improved based on the suggested changes during the first revision stage. Therefore, the authors are requested again to carefully revise the manuscript based on the points raised:

The currently explained contributions of the paper are not adequate according to the level of the journal. For a conference paper, the contributions are enough; however, the contributions are not adequate to be considered for publication in a reputed Applied Science journal. Therefore, the authors are suggested to elaborate the main contributions of the paper in a detailed way.

About the literature review, instead of adding the previously suggested references, the authors have self-cited their own references, which actually is not professional. Therefore, the authors must add the previously suggested three references.

Some of the reference’s description is not up to the mark. Therefore, please revisit the descriptions.

The details of data collection are still lacking proper details and descriptions. Therefore, please provide all the missing information.

What kind of python libraries the authors are talking about in terms of feature scaling? Please provide all the details related to feature scaling because it is an important stage of model designing.

About the reason of not using deep learning approaches, the authors are suggested to add comments related to data issue in deep learning because the given reason is not appropriate.

In the below previously suggested point, the authors were suggested to compare the result with the other existing research papers, but they have answered completely different. Therefore, please provide a better comparison.

“The comparison is missing in the paper. Therefore, please provide some comparison between the utilized method and the existing approaches to highlight the main novelty of the paper.

The authors have not calculated any performance parameters of regression, such as MAE, etc. please add the missing performance parameters.

Readership appeal is still moderate. Therefore, please provide more pictorial information.

 It is quite difficult to reproduce the study. Therefore, please provide the values of all parameters set during the simulations. (Here, provide all the used parameters, such as hyperparameters, etc.)

The currently explained limitations are not adequate. Therefore, please improve it.

Minor editing of English language required.

Author Response

To the Reviewer 

We are pleased to receive your comments regarding our manuscript entitled “A Study on Analysis of Production Data of Feed Crops and Vulnerability to Climate Impacts According to Climate Change in South Korea”.

We have revised our manuscript again.

One of the references you suggested previously has been added. Additional reference was decided by consensus of the authors.

Data collection was created by examining reports and experimental cultivation results from the National Institute of Animal Science, as described previous version. It was provided by the Korea Institute of Animal Science and Technology, and this study was the result of joint research with the Korea Institute of Animal Science.

We do not think that it was necessary adding program code in the paper. Therefore, we added a part of Python program as appendix.

The climate data crawler is installed and used by the Korea Institute of Animal Science and Technology.

Climate data contains multicollinearity. Therefore, it was found to be appropriate to apply the LASSO model or the RIDGE model.

Lasso Regression is a popular type of regularized linear regression that includes an L1 penalty. This has the effect of shrinking the coefficients for those input variables that do not contribute much to the prediction task. This penalty allows some coefficient values to go to the value of zero, allowing input variables to be effectively removed from the model, providing a type of automatic feature selection.

We focused on deriving the climate factors that most influence forage crop productivity. Therefore, we conducted a comparative analysis of various data sets and various regression models and derived better R2 score. So, IRG was found to be most influenced by the elements MINTJAN, PREOCT, and GDDFJTA, and grass was found to be greatly influenced by PREDAUG, GDDTOTAL, MAXJUL, MAXAUG and DDAYS.

R2 was used to compare the performance of regression models. There are many factors to evaluate accuracy or error rate such as MAE, MSE, MAPE, correlation coefficient or determinant coefficient R2. In our study, we thought R2 is meaningful.

Reproduction is possible using the Python program source and dataset.

We appreciate your helpful comments, which we feel have improved our manuscript.

Round 3

Reviewer 2 Report

The manuscript was much improved as suggested. It can be accepted for publication.